# Cog-Rethinker: Hierarchical Metacognitive Reinforcement Learning for LLM Reasoning

## Abstract

Contemporary progress in large language models (LLMs) has revealed notable inferential capacities via reinforcement learning (RL) employing rule-based reward, facilitating the development of O1 and R1-like reasoning models. Directly training from base models with RL is called zero-RL. However, previous works rely upon activating LLMs' inherent capacities through fixed prompt templates. This strategy introduces substantial sampling inefficiencies for weak LLMs, as the majority of problems generate invalid outputs during accuracy-driven filtration in reasoning tasks, which causes a waste of samples. To solve this issue, we propose Cog-Rethinker, a novel hierarchical metacognitive RL framework for LLM reasoning. Our Cog-Rethinker mainly focuses on the rollout procedure in RL training. After the direct rollout, our Cog-Rethinker improves sample utilization in a hierarchical metacognitive two-stage framework. By leveraging human cognition during solving problems, firstly, it prompts policy to decompose zero-accuracy problems into subproblems to produce final reasoning results. Secondly, with zero-accuracy problems in previous rollout stage, it further prompts policy to refine these answers by referencing previous wrong solutions. Moreover, to enable cold-start of the two new reasoning patterns and maintain train-test consistency across prompt templates, our Cog-Rethinker applies supervised fine-tuning on the policy using correct samples of the two stages with direct rollout template. Experimental results demonstrate Cog-Rethinker's superior performance on various mathematical reasoning benchmarks, we also analyzed its improved sample efficiency that accelerates convergence compared to baseline methods.

## 1 Introduction

Recent developments in Large Language Models (LLMs) have exhibited significant advancements in inferential capacities, achieving unprecedented accuracy in complex reasoning challenges and even surpassing human performance in specialized disciplines. Prominent examples including OpenAI's O1 (Jaech et al., 2024), Google's Gemini-2.0 (Google, 2024), DeepSeek-R1 (Guo et al., 2025), and Qwen-QwQ (Team, 2024) demonstrate these improvements through their capacity to replicate human-like systematic reasoning methodologies. Performance optimization is achieved through deliberate temporal resource allocation during inference phases. Despite these breakthroughs, it is still challenging when addressing exceptionally demanding tasks such as mathematical reasoning (Li et al., 2024; He et al., 2024) and program synthesis (Jain et al., 2024), which necessitates exploration of expansive solution spaces and meticulous execution of intricate reasoning steps.

Contemporary investigations have prioritized advancing LLMs' sophisticated reasoning capacities through inference-phase optimization strategies. The zero-RL framework (Guo et al., 2025; Zeng et al., 2025; Liu et al., 2025) has emerged as particularly effective, implementing RL on base model by leveraging their own rollouts. Despite empirical validation, zero-RL exhibits inherent limitations imposed by the foundational competency profile of base LLMs (Zhao et al., 2025), primarily reinforcing pre-existing patterns instead of novel cognitive capacities. Recent studies (Gandhi et al., 2025; Zhang et al., 2025) have demonstrated this limitation, revealing that models such as Llama 3.2 (Meta AI Team, 2024) quickly reach performance plateaus in zero-RL training due to the absence

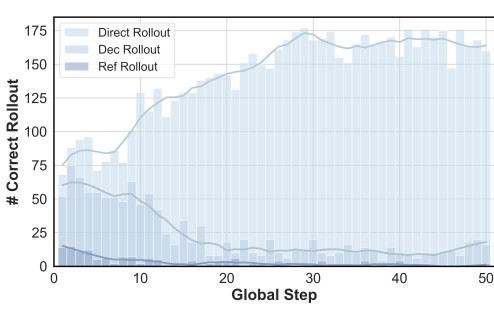 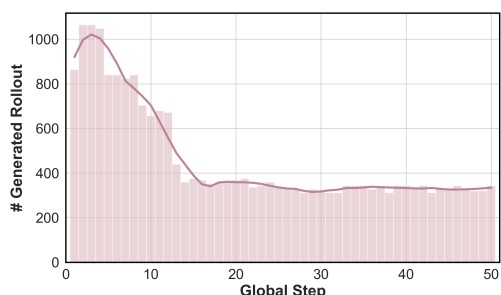

(a) Problem accuracy across training steps  (b) Sample generation statistics during training

Figure 1: The case study of our Cog-Rethinker. The Dec Rollout and Ref Rollout denote the policy generation by problem decomposition and answer reflection, respectively. Our Cog-Rethinker can generate more correct samples especially at the beginning of training procedure.

of fundamental cognitive mechanisms. Existing approaches that depend on fixed prompt templates exacerbate these issues, while the accuracy filter leads to significant sample waste during early training stages. However, the incorporation of negative samples through more principled designs (Xiong et al., 2025), can enhance the performance of reasoning models, particularly for weaker base models in zero-RL. Existing research in cognitive science (Thagard, 2013; Endsley et al., 2007) shows that problem solving can benefit from cognition. This provokes a crucial research question:

*How can we enable LLMs to acquire reasoning behaviors for the negative samples that fully transcend their initial cognitive boundaries?*

In this work, we propose Cog-Rethinker, a novel hierarchical metacognitive reinforcement learning framework designed to enhance LLM reasoning capabilities. Unlike existing approaches that rely solely on direct rollout, our Cog-Rethinker introduces two hierarchical metacognitive rollout stages. In the first stage, with the zero-accuracy problem after the direct rollout, our Cog-Rethinker incentivize policy to decompose the problem into manageable subproblems with a provided meta demonstration for sequential solving. But, with the policy reasoning ability improving during training, the simple fixed demonstration cannot fully motivate the policy to provide correct decomposition with hard problems. To alleviate this, we implement a memory buffer to store the correct decomposition samples generated by the policy itself. With demonstrations dynamically retrieved based on problem similarity in the decomposition template in the following rollout. In the second stage, with the problems of zero-accuracy in the first stage, our Cog-Rethinker prompts policy to revise incorrect solutions by referencing previous wrong solutions in a structured reflection template. Moreover, to maintain train-test consistency and inject new reasoning patterns for cold-start scenarios, we restore all samples in the replay buffer to their original prompt templates and apply supervised fine-tuning (SFT) to the policy using correct samples from the two stages. Our Cog-Rethinker significantly accelerates policy convergence while requiring fewer training samples. We conduct a training visualization of our Cog-Rethinker on the Qwen2.5-1.5B-Base model in Figure 1, the two rollout stages lead to a significant increase in positive sample generation early in training and a consequent major gain in sample utilization efficiency. Our main contribution is summarized as follows:

- We propose Cog-Rethinker, a novel hierarchical metacognitive reinforcement learning framework that introduces two additional rollout stages – decomposition and reflection rollout, which significantly enhance sample utilization efficiency in LLM reasoning training.
- To ensure stable training and testing dynamics, we develop an adaptive metacognitive buffer for metacognative rollout and apply SFT to policy with correct samples in two stages.
- Through experiments across multiple reasoning benchmarks, we demonstrate that Cog-Rethinker achieves better performance while requiring fewer samples compared to existing approaches.

## 2 RELATED WORK

**Reinforcement Learning with Verifiable Reward (RLVR).** Leveraging rule-based verification for reward computation has become increasingly prevalent in enhancing LLMs' reasoning capabilities

(Lambert et al., 2024; Guo et al., 2025; Team et al., 2025). Unlike preference-based approaches that require human feedback collection (Christiano et al., 2017; Ouyang et al., 2022; Bai et al., 2022; Song et al., 2023), RLVR employs deterministic verification functions, most commonly answer matching in mathematical domains to generate binary reward signals that guide model optimization (Guo et al., 2025; Team et al., 2025; Zeng et al., 2025; Xie et al., 2025). The PPO (Schulman et al., 2017) algorithm is the most commonly used reinforcement learning training algorithm. However, when applied to the field of LLMs training, the PPO algorithm often suffers from excessively high resource consumption. As a result, new algorithms have recently been proposed from the perspectives of resource efficiency and training acceleration, including GRPO (Shao et al., 2024), Reinforce++ (Hu et al., 2025a), and other similar variants (Yu et al., 2025; Lin et al., 2025; Kazemnejad et al., 2024; Yuan et al., 2025; Liu et al., 2025). Recent industry breakthroughs like OpenAI o1 (OpenAI et al., 2024) and DeepSeek-R1 (Guo et al., 2025) have demonstrated RLVR's potential to develop models with superior reasoning patterns.

**Inference Scaling for LLM Reasoning.** The auto-regressive nature of LLMs necessitates increased token generation for complex problem-solving. Foundational work like Chain-of-Thought (CoT) (Wei et al., 2022) introduced step-by-step prompting to decompose reasoning tasks, significantly improving performance. Subsequent approaches including Tree-of-Thoughts (ToT) (Yao et al., 2024) and Graph-of-Thoughts (GoT) (Besta et al., 2024) expanded the solution space through structured reasoning pathways. Recent theoretical advances (Wu et al., 2024; Snell et al., 2024) have established inference scaling laws that quantify the trade-offs between token generation and inference strategies. Current methods employ various techniques: majority voting and best-of-N sampling (Wang et al., 2022; Li et al., 2023) generate multiple solutions for optimal selection, while Monte Carlo Tree Search (MCTS) approaches (Zhang et al., 2024; Liu et al., 2024; Choi et al., 2023; Zhou et al., 2023) enhance accuracy through extensive computation. Process Reward Models (PRMs) (Setlur et al., 2024; Snell et al., 2024; Lightman et al., 2023; Wang et al., 2024) have proven particularly effective for complex reasoning by selecting high-quality reasoning paths. Modern methods like Bootstrapped Thought (BoT) (Yang et al., 2024b) leverage historical reasoning templates to guide exploration, though the exploration-exploitation balance in template-based approaches (Tang et al., 2024; Setlur et al., 2024) remains unresolved. Our Cog-Rethinker advances this frontier through hierarchical metacognitive reinforcement learning, combining template-augmented reasoning with enhanced sample efficiency to achieve superior accuracy.

## 3 PRELIMINARIES

**Cognitive Engineering.** As demonstrated in Xia et al. (2025), cognitive engineering marks a paradigm shift in AI development. To analyze this emerging discipline, we employ the DIKW (Data-Information-Knowledge-Wisdom) hierarchy (Zeleny, 1987; Ackoff, 1989) as a theoretical framework, examining how cognitive engineering facilitates the transition from knowledge to wisdom. The key distinction between cognitive engineering and traditional LLM development approaches lies in their fundamental methodologies. Cognitive engineering specifically emulates human thought processes, directly targeting the cognitive attributes of the wisdom level.

**Decouple Clip and Dynamic Sampling Policy Optimization (DAPO).** DAPO (Yu et al., 2025) represents an improved version of the GRPO (Shao et al., 2024) algorithm. During practical training, DAPO samples a group of outputs $\{o_i\}_{i=1}^{G}$ for each question-answer pair $(q, a)$ and optimizes the policy through the following objective function:

$$
\begin{aligned}
\mathcal{L}_{\text{DAPO}}(\theta) = \quad & -\mathbb{E}_{(q,a)\sim\mathcal{D}, \{o_i\}_{i=1}^{G}\sim\pi_{\theta_{\text{old}}}(\cdot|q)} \\
& \left[ \frac{1}{\sum_{i=1}^{G}|o_i|} \sum_{i=1}^{G}\sum_{t=1}^{|o_i|} \min\left( r_{i,t}(\theta)\hat{A}_{i,t},\ \text{clip}\left(r_{i,t}(\theta), 1-\varepsilon_{\text{low}}, 1+\varepsilon_{\text{high}}\right)\hat{A}_{i,t} \right) \right] \quad (1) \\
\text{s.t.} \quad & 0 < \left| \{o_i \mid \texttt{is\_equivalent}(a, o_i)\} \right| < G,
\end{aligned}
$$

where,

$$
r_{i,t}(\theta) = \frac{\pi_\theta(o_{i,t} \mid q, o_{i,<t})}{\pi_{\theta_{\text{old}}}(o_{i,t} \mid q, o_{i,<t})}, \quad \hat{A}_{i,t} = \frac{r_i - \text{mean}(\{r_i\}_{i=1}^{G})}{\text{std}(\{r_i\}_{i=1}^{G})}.
$$

Since reward models often suffer from reward hacking (Amodei et al., 2016; Everitt et al., 2017; 2021; Gao et al., 2023), in mathematical reasoning tasks, a simpler rule-based matching approach is typically employed to determine whether the final answer is correct, providing a binary reward signal. Specifically, given a question-answer pair $(q, a)$ and an output $o$, the binary reward model is typically defined as,

$$R(a, o; x) = \begin{cases} 1, & \text{is\_equivalent}(a, o), \\ -1, & \text{otherwise.} \end{cases} \tag{2}$$

The use of this binary reward function in RL enhances training stability and reliability by substantially reducing vulnerabilities to reward hacking.

## 4 METHODOLOGY

In this section, to easily understand the overall structure of our Cog-Rethinker, we present the detail visualization in Figure 2. In a high level, our Cog-Rethinker mainly focus on the accuracy filter-based rollout stage in RL training, thus, we directly introduce it from three aspects, the decomposition rollout, the reflection rollout and the policy training.

### 4.1 DECOMPOSITION ROLLOUT

In our Cog-Rethinker, with the zero-accuracy problem after the direct rollout, we apply the decomposition rollout in the first stage. When tackling complex reasoning tasks, human problem-solvers frequently resort to decomposition techniques and analogical reasoning strategies for particularly difficult problems (Landauer et al., 1997). Motivated by this, within the domain of LLMs, existing research in LLMs has provided empirical validation for the effectiveness of such decomposition methods (Xue et al., 2024; Jiang et al., 2022; Zhao et al., 2023; Zhou et al., 2025).

For a given complex mathematical reasoning problem $Q$, how to incentivize policy to decompose the problem in our desired manner remains a challenge. Existing works (Xue et al., 2024; Sarangi et al., 2025) show that when provided with specific decomposition demonstrations, LLMs are capable of breaking down the problem in accordance with the expected format. Therefore, we maintain a metacognitive buffer $\mathcal{M}$ of decomposition demonstrations and pre-construct a set of problem decomposition demonstrations. These examples serve as the reference for the model to learn decomposition patterns and improve its ability to break down complex problems.

Specifically, we retrieve the most similar problem $\hat{Q}$ from the decomposition example metacognitive buffer based on problem similarity to assist in the decomposition process,

$$\{\hat{Q}, \{(\hat{q}_i, \hat{a}_i)\}_{i=1}^k, \hat{A}\} = \underset{Q_i \in \mathcal{M}}{\arg\max} \, \text{sim}(Q, Q_i). \tag{3}$$

Here, we utilize BM25 (Robertson et al., 2009) for similarity-based retrieval.

$$\text{sim}(Q, Q_i) = \sum_{w \in Q} \text{IDF}(w) \cdot \frac{f_{w, Q_i} \cdot (k+1)}{f_{w, Q_i} + k \cdot (1 - b + b \cdot \frac{|Q_i|}{\text{avg}_{\mathcal{M}}})}, \tag{4}$$

where $\text{IDF}(w)$ (Spärck Jones et al., 1998) measures how important a word $w$ is in the question $Q$, downweighting common terms and highlighting rare, meaningful ones. $f_{w, Q_i}$ denotes the frequency of the word $w$ in $Q_i$, $|Q_i|$ represents the length of the query $Q_i$, $\text{avg}_{\mathcal{M}}$ is the average question length in buffer $\mathcal{M}$, and $k$ and $b$ are hyperparameters. In our experiments, we set $k = 1.2$ and $b = 0.75$.

Compared to other similarity retrieval algorithms, BM25 demonstrates superior performance in handling text length variations, particularly for long-form responses and extended sequences. Additionally, its lightweight computational cost makes it suitable for integration into RL training. By leveraging this metacognitive strategy, our Cog-Rethinker enhances the policy's ability to break down intricate problems into manageable sub-tasks.

After obtaining the most similar question $\hat{Q}$ to the original question $Q$, we prompt the policy to perform an explicit problem decomposition process. Specifically, we input the original question $Q$, the similar question $\hat{Q}$, along with its decomposition and solution process $\{(\hat{q}_i, \hat{a}_i)\}_{i=1}^k$, as well as the final answer $\hat{A}$ into the policy, enabling it to carry out the corresponding decomposition.

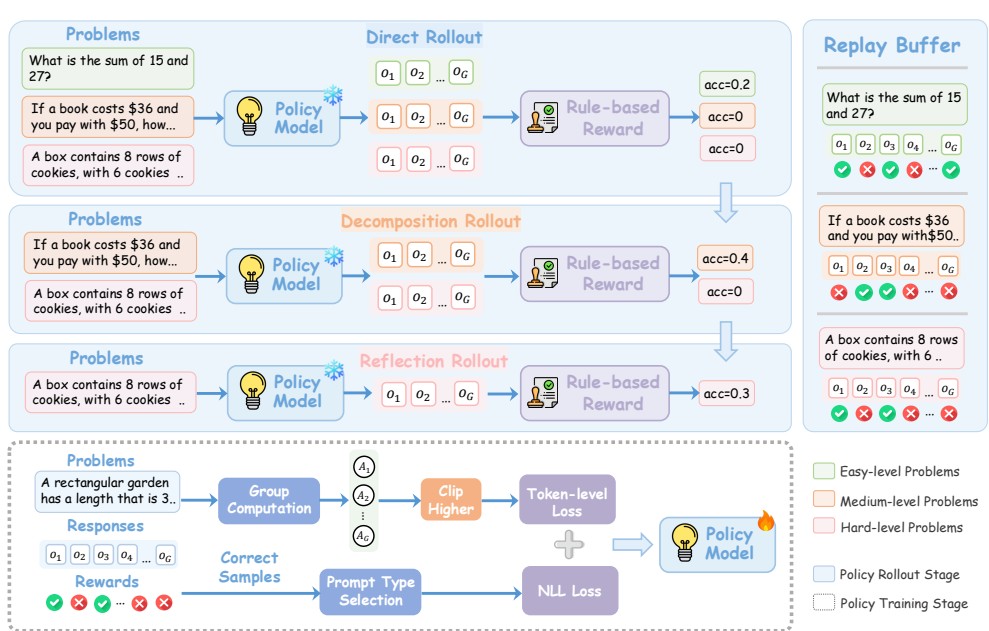

Figure 2: Overall procedure of our Cog-Rethinker. The upper is the whole rollout stage for different difficulty problems, the lower is the training procedure with token-level policy gradient loss of DAPO and NLL loss of SFT.

After that, we guide the policy to sequentially solve these subproblems, generating corresponding question-answer pairs $\{(q_i, a_i)\}_{i=1}^n$. Based on these subproblem-answer pairs, we finally prompt the policy to produce the solution to the original problem,

$$\{q_i\}_{i=1}^n = Decompose(\pi_\theta, Q, \hat{Q}, \{(\hat{q}_i, \hat{a}_i)\}_{i=1}^k, \hat{A}), \quad A \sim \pi_\theta(\cdot \mid Q, \{(q_i, a_i)\}_{i=1}^n), \quad (5)$$

where $\pi_\theta$ represents the policy.

However, predefining decomposition demonstrations is time-consuming and labor-intensive, which limits the scalability and adaptability. To overcome this bottleneck, we propose an automated approach for generating diverse decomposition demonstrations, ensuring sustained and efficient utilization of the decomposition process. Specifically, our Cog-Rethinker dynamically updates the metacognitive buffer $\mathcal{M}$ by integrating the questions that are successfully solved after decomposition but previously unresolved through direct response:

$$\mathcal{M} \leftarrow \mathcal{M} \cup (Q, \{(q_i, a_i)\}_{i=1}^k, A), \quad \text{if } R = 1 \quad (6)$$

where $R = 1$ indicates that the policy generated the correct answer. Through dynamic augmentation of the metacognitive buffer with additional decomposition demonstrations, we increase the diversity of available decomposition strategies. Furthermore, the buffer is designed with maximum capacity and a first-in-first-out (FIFO) structure to better align with the current policy's capabilities during RL training, thereby providing higher-quality options.

## 4.2 REFLECTION ROLLOUT

In our Cog-Rethinker, we apply reflection rollout in the second stage with the problems that is zero-accuracy filtered by the decomposition stage. It incentivizes the policy to revise the answer with the previous wrong answer as the metacognition, which is called the reflection rollout.

Specifically, given a problem $Q$, its corresponding decomposition and solution steps $\{(q_i, a_i)\}_{i=1}^n$, and the final wrong answer $A$, our Cog-Rethinker prompts policy to systematically re-evaluate and correct the reasoning process:

$$(Q, \{(q_i', a_i')\}_{i=1}^n, A') = Reflect(\pi_\theta, Q, \{(q_i, a_i)\}_{i=1}^n, A) \quad (7)$$

where $A'$ and $(q', a')$ are the answer and solution steps generated by the reflection rollout, we aim to enable the policy to conduct fine-grained reflection on the entire reasoning process, which involves

Figure 3: Different prompt templates of our Cog-Rethinker during whole rollout stage.

two key aspects: (1) revising the sub-questions $\{q_i\}_{i=1}^n$ and (2) correcting the resolutions $\{a_i\}_{i=1}^n$ to these sub-questions. Both inadequate problem decomposition and erroneous sub-question responses can hinder the generation for correct answer, necessitating meticulous refinement.

## 4.3 POLICY TRAINING

Following direct rollout and above two rollout stages, all samples with accuracy scores between 0 and 1 are collected into the replay buffer $\mathcal{B}$ for policy training. However, these samples present a critical inconsistency: the prompt templates differ between training and testing phases. During rollout stage of RL training, three distinct prompt templates are employed, while testing utilizes only the direct rollout template.

To alleviate this and inject the new reasoning patterns into policy training, our Cog-Rethinker modifies the token-level policy gradient loss in Eq. (1) by incorporating clip-higher regularization. Furthermore, to integrate the two new reasoning patterns introduced during policy rollout, we implement Supervised Fine-Tuning (SFT) (Chu et al., 2025) alongside RL training. This hybrid approach specifically targets correct problems generated through decomposition and reflection rollout, systematically transferring these reasoning capabilities into the policy's direct response generation. Specifically, we incorporate the following additional loss function,

$$\mathcal{L}_{\text{SFT}}(\theta) = - \mathbb{E}_{\substack{(Q,\{(q_i,a_i)\}_{i=1}^k, A, R) \sim \mathcal{B} \\ Q \in \{\text{Decompose, Reflect}\} \& R = 1}} \left[ \log \pi_\theta \big( (\{(q_i, a_i)\}_{i=1}^k, A) \mid Q \big) \right], \tag{8}$$

Specially, we replace the prompt template of $Q \in \{\text{Decompose, Reflect}\} \& R = 1$ into the direct rollout template to keep the training testing consistency.

Ultimately, we obtain the final loss function of Cog-Rethinker as follows:

$$\mathcal{L}_{\text{Cog-Rethinker}}(\theta) = \mathcal{L}_{\text{DAPO}}(\theta) + \lambda \mathcal{L}_{\text{SFT}}(\theta). \tag{9}$$

where $\lambda$ is the hyperparameter to control the trade-off between RL and SFT training.

To better understand the effectiveness of our Cog-Rethinker, we conduct Theorem 1 to analyze the convergence rate of three different rollout stages, Direct Rollout (DR), Decomposition Rollout (DecR) and Reflection Rollout (RefR), respectively.

**Theorem 1** (Convergence Rate across Stages). *Let $m \in \{\text{DR, DecR, RefR}\}$ index the rollout stages of Cog-Rethinker. Assume (i) horizon $H$ is finite and rewards $r_m(\cdot, \cdot) \in [0, 1]$; (ii) for DecR, the problem is decomposed into sub-problems of horizon $H' < H$; (iii) for RefR, reflection is performed on a sub-tree of horizon $H'' \leq \gamma H'$ with $\gamma \in (0, 1)$. Then the policy-gradient estimator,*

$$g_m(\theta) = \sum_{t=0}^{H_m - 1} \nabla_\theta \log \pi_\theta(a_t | q_t) \, G_{t,k}, \quad G_{t,m} = \sum_{u=t}^{H_m - 1} r_m(q_u, a_u)$$

*satisfies*

$$\text{Var}(g_{\text{RefR}}) \ \leq \ \gamma(1 - \eta) \, \text{Var}(g_{\text{DecR}}) \ < \ \text{Var}(g_{\text{DecR}}) \ < \ \text{Var}(g_{\text{DR}}),$$

Table 1: Overall accuracy performance on various reasoning benchmarks. The best and second best results are in **bold** and underlined.

| Method | GSM8K | MATH-500 | AIME 24 | AIME 25 | AMC 2023 | Gaokao 2023en | Minerva | GPQA-diamond | Olympiad |
|---|---|---|---|---|---|---|---|---|---|
| Qwen2.5-1.5B-Base | 37.98 | 21.60 | 3.33 | 0.00 | 15.00 | 14.81 | 4.04 | 5.65 | 5.33 |
| PPO | 67.78 | 41.20 | 0.00 | 0.00 | 28.50 | 38.81 | 15.44 | 22.31 | 17.67 |
| GRPO | 69.67 | 42.00 | **6.67** | 0.00 | 32.50 | 37.92 | 15.44 | 20.11 | 16.00 |
| Reinforce++ | 63.15 | 44.20 | 0.00 | 0.00 | 30.00 | 31.43 | 14.71 | 24.36 | 18.07 |
| BODF | 74.07 | 51.20 | 3.33 | 0.00 | 42.50 | 44.16 | 15.07 | 20.07 | 18.96 |
| DAPO | **77.56** | 56.00 | **6.67** | 0.00 | **47.50** | 42.34 | 16.54 | 19.53 | **22.22** |
| **Cog-Rethinker** | 77.51 | **59.00** | **6.67** | **6.67** | **47.50** | **44.94** | **17.65** | **24.40** | **22.22** |
| Qwen2.5-7B-Base | 58.91 | 41.60 | 6.67 | 0.00 | 52.50 | 29.87 | 11.40 | 18.55 | 14.67 |
| PPO | 90.55 | 76.40 | 20.00 | 16.67 | 70.00 | 61.82 | 31.62 | 23.85 | 40.78 |
| GRPO | 92.12 | 78.40 | **26.67** | 20.00 | 72.50 | 61.56 | **33.09** | 33.24 | 41.48 |
| Reinforce++ | 91.36 | 78.20 | 20.00 | 23.33 | 70.00 | 61.82 | 31.62 | 23.85 | 42.22 |
| BODF | 91.58 | 74.40 | 20.00 | 16.67 | 62.50 | 60.00 | 31.25 | 26.76 | 37.78 |
| DAPO | 92.21 | 79.40 | **26.67** | **26.67** | 69.00 | 63.22 | 31.88 | 34.65 | 42.44 |
| **Cog-Rethinker** | **93.32** | **80.60** | **26.67** | **26.67** | **73.50** | **65.52** | 32.98 | **36.42** | **44.22** |

Table 2: Ablation study on various mathematical reasoning benchmarks. The best and second best results are in **bold** and underlined.

| Method | GSM8K | MATH-500 | AIME 24 | AIME 25 | AMC 2023 | Gaokao 2023en | Minerva | GPQA-diamond | Olympiad |
|---|---|---|---|---|---|---|---|---|---|
| **Cog-Rethinker-1.5B** | **77.51** | **59.00** | **6.67** | **6.67** | 47.50 | **44.94** | **17.65** | **24.40** | 22.22 |
| Cog-Rethinker w/o SFT | 72.25 | 51.40 | 3.33 | 0.00 | **50.00** | 37.66 | 15.81 | 23.86 | 20.89 |
| Cog-Rethinker w/o MB | 75.89 | 55.60 | 3.33 | 0.00 | **50.00** | 43.90 | 14.71 | 23.86 | **23.26** |
| Cog-Rethinker w/o RefR | 74.45 | 54.80 | 3.33 | 3.33 | 42.50 | 41.82 | 17.28 | 19.09 | 18.67 |
| **Cog-Rethinker-7B** | **93.32** | **80.60** | **26.67** | **26.67** | **73.50** | **65.52** | **32.98** | **36.42** | **44.22** |
| Cog-Rethinker w/o SFT | 91.66 | 77.00 | 16.67 | 16.67 | 70.50 | 61.04 | 29.04 | 31.43 | 39.41 |
| Cog-Rethinker w/o MB | 92.34 | 78.00 | 20.00 | 13.33 | 65.00 | 60.00 | 30.88 | 30.88 | 38.67 |
| Cog-Rethinker w/o RefR | 92.12 | 80.40 | 20.00 | 10.00 | 65.00 | 59.48 | 31.62 | 31.44 | 42.07 |

*where $\eta \in (0,1)$ is the variance-reduction factor induced by importance-sampling the error sub-tree,* $\mathrm{Var}(\cdot)$ *represents the variance Consequently, for target accuracy $\epsilon > 0$,*

$$T_{\mathrm{RefR}}(\epsilon) \; < \; T_{\mathrm{DecR}}(\epsilon) \; < \; T_{\mathrm{DR}}(\epsilon).$$

*where $T(\cdot)$ represents the iteration complexity.*

We provide the related proof in Appendix A. Thus, our Cog-Rethinker achieves better convergence than the direct rollout method given the same number of rollouts on negative samples.

## 5 EXPERIMENTS

In this section, we present comprehensive experimental results and analysis of our Cog-Rethinker against other baselines. Our experiments focus on the following research questions:

- **RQ1:** Can our Cog-Rethinker outperforms all the baseline method across various benchmarks?
- **RQ2:** How each part of our Cog-Rethinker affects the model performance?
- **RQ3:** Can our Cog-Rethinker improves the sample efficiency during training?

**Training Details.** We initialize both our policy and critic networks with Qwen-2.5-base models (1.5B and 7B) (Yang et al., 2024a), where value head is random initialized from $\mathcal{U}(-\sqrt{5}, \sqrt{5})$ with no bias term. For policy networks, we employ AdamW optimizer with $\beta = [0.9, 0.95]$ without weight decay. The learning rate is set to $1 \times 10^{-6}$ for the policy. The learning rate scheduler are both constant learning rate with linear warm-up of 50 optimizer steps. We employ sample packing during training. We use orz-math-127k as the training dataset (Hu et al., 2025b), also we develop our code based on VeRL (Sheng et al., 2024). Each generation step contains 128 unique prompts sampled from the dataset, and generates 64 responses per prompt with temperature and top-p both set to 1.0. To maintain training stability, we keep the size of the replay buffer as 128 unique prompts until it is satisfied with the accuracy filter.

**Evaluation Benchmarks.** To evaluate the complex reasoning capabilities, we choose a broad set of challenging reasoning benchmarks, including GSM8K (Cobbe et al., 2021), MATH500 (Hendrycks

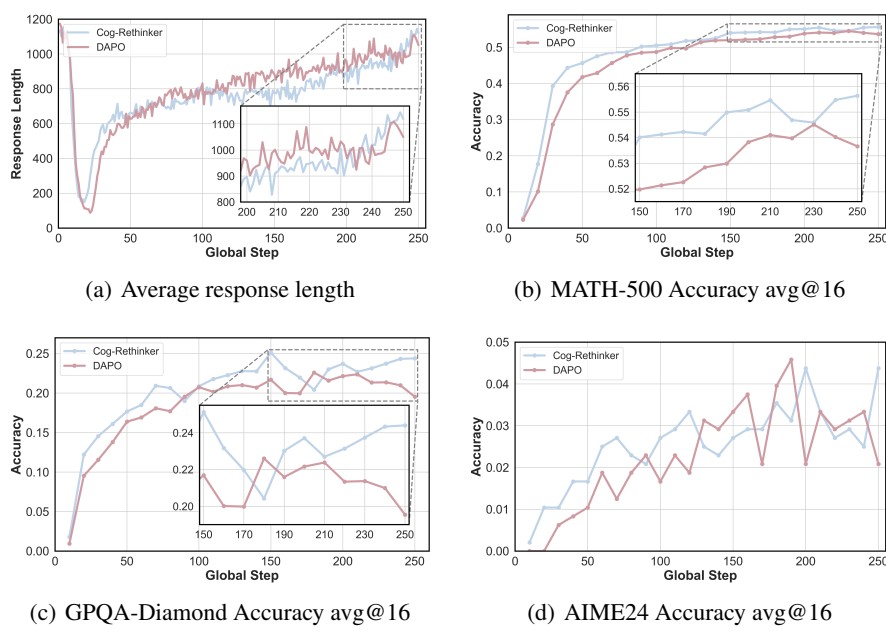

(a) Average response length

(b) MATH-500 Accuracy avg@16

(c) GPQA-Diamond Accuracy avg@16

(d) AIME24 Accuracy avg@16

Figure 4: Training comparison between our Cog-Rethinker and DAPO.

et al., 2021), AIME 2024 and 2025 (Li et al., 2024), AMC 2023 (Li et al., 2024), Gaokao 2023en (Liao et al., 2024), GPQA-diamond (Rein et al., 2024), Minera (Lewkowycz et al., 2022) and OlympiadBench (He et al., 2024). These benchmarks comprehensively evaluate mathematical reasoning capabilities, and they are all competition-level and Olympic-level problems. Moreover, AIME 2024, 2025 and AMC 2023 are highly challenging competition benchmarks, the results are through majority voting across 16 runs.

**Baselines.** To demonstrate the reasoning ability of our Cog-Rethinker, we compare it with many strong baseline methods: PPO (Schulman et al., 2017), GRPO (Shao et al., 2024), Reinforce++ (Hu, 2025), BODF (Bae et al., 2025) and DAPO (Yu et al., 2025). Specifically, PPO, GRPO and Reinforce++ are the commonly used methods for reproducing the O1 and R1-like reasoning models. BODF is the extension of accuracy filtering-based methods (Yu et al., 2025; Cui et al., 2025) by designing the balanced filtering with theoretical guarantees. DAPO leverages the dynamic sampling to improve the training efficiency and stability. Additionally, we choose the accuracy rate of rollout samples between 0.3 and 0.7 in BODF optimization.

## 5.1 OVERALL PERFORMANCE (RQ1)

Table 1 shows the final results of our Cog-Rethinker with a comprehensive comparison to SOTA reasoning methods. We find that our Cog-Rethinker consistently outperforms the baselines on most challenging mathematical benchmarks across the 1.5B and 7B size base models. More specifically, over the results of 1.5B model, our Cog-Rethinker achieves highest score in MATH-500, surpassing the nearest competitor DAPO by 3.00%, and demonstrates exceptional adaptability in AIME 24 and AMC 2023, outperforming all baselines. Notably, Cog-Rethinker uniquely solves AIME 25 where all other methods score 0.00%, highlighting its capacity for highly challenging tasks. While narrowly trailing DAPO in GSM8K. Over the results of 7B models, our Cog-Rethinker stands out as the top-performing method, achieving the highest scores in most datasets. It leads with 93.32% on GSM8K, 80.60% on MATH-500, 26.67% on both AIME 24 and 25, 73.50% on AMC 2023, 65.52% on Gaokao 2023en, 36.42% on GPQA-diamond, and 44.22% on Olympiad, demonstrating consistent superiority. Other methods like GRPO, DAPO, and Reinforce++ show competitive results but fall short of our Cog-Rethinker's performance. For instance, DAPO scores 92.21% on GSM8K and 26.67% on AIME 25, while GRPO achieves 78.40% on MATH-500, both trailing behind our Cog-Rethinker. The base model, Qwen2.5-7B-Base, performs the weakest, highlighting the significant improvements brought by advanced techniques. Our Cog-Rethinker's dominance across diverse and complex tasks underscores its effectiveness in tackling challenging problems.

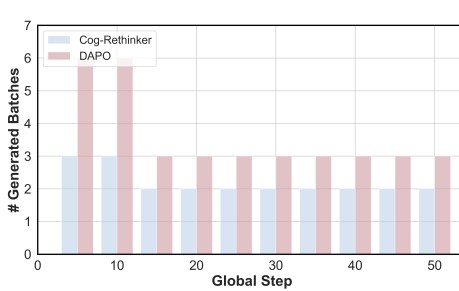 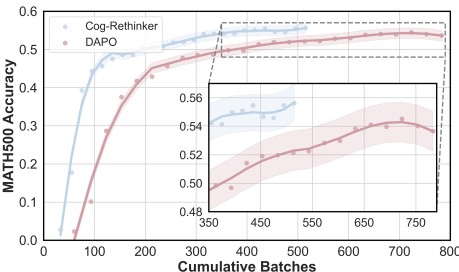

(a) Generated Batches Over Training Steps     (b) Cumulative Batches vs MATH500 Accuracy

Figure 5: Sample utilization efficiency analysis between our Cog-Rethinker and DAPO.

## 5.2 ABLATION STUDY (RQ2)

In this section, we conduct experiments to verify the effectiveness of each part in our Cog-Rethinker. Specially, we sequentially remove the SFT for correct sample of decomposition and reflection, metacognitive buffer of decomposition rollout (MB) and reflection rollout (RefR) to test the newly trained policies. We make three variants (Cog-Rethinker w/o SFT, Cog-Rethinker w/o MB, Cog-Rethinker w/o RefR), the results are shown in Table 2. From the results in Table 2 on both 1.5B and 7B models, we can see that, while removing any component degrades results. SFT removal causes the steepest decline, with GSM8K dropping to 72.25% of 1.5B model, underscoring its role in knowledge injection for the base model to cold start. Ablating MB reduces consistency, causing MATH-500 falling to 78.00% of the 7B model's performance, highlighting its importance for the decomposition rollout. The removal of RefR weakens performance, with Olympiad scores dropping to 18.67 for the 1.5B model, proving its importance in optimizing complex tasks. The 1.5B variant's significantly weaker performance confirms the advantages of scale, shows that our Cog-Rethinker improve the training of weaker models.

## 5.3 TRAINING EFFICIENCY (RQ3)

In this section, we visualize the training procedure of our Cog-Rethinker compared with DAPO to demonstrate its effectiveness. Figure 4(a) shows that Cog-Rethinker achieves shorter stabilized response lengths, indicating more efficient output refinement. Figures 4(b) and (c) reveal that our method maintains consistent performance advantages over DAPO, suggesting superior convergence properties. Finally, Figure 4(d) demonstrates that Cog-Rethinker continuously improves performance on challenging tasks, being competitive with DAPO throughout training. To further analyze sample efficiency, we conduct experiments comparing the relationship between training samples used and final model performance, with results presented in Figure 5. Figure 5(a) demonstrates that our Cog-Rethinker is capable of obtaining more valid training samples than DAPO, reduces the batch generation overhead before both methods reach stability. Figure 5(b) reveals a positive correlation between cumulative batches and MATH500 accuracy, with Cog-Rethinker exhibiting superior sample efficiency throughout the training dynamics, which also confirms the analysis in Theorem 1.

## 6 CONCLUSION

In this paper, we propose Cog-Rethinker, a hierarchical metacognitive reinforcement learning framework that advances beyond zero-RL through two key mechanisms: (1) hierarchical integration of problem decomposition and reflection in rollout stage to transcend initial cognitive constraints, and (2) adaptive memory for demonstration retrieval of prompt templates and combined with SFT to cold start and keep the train-test consistency. Empirical results show state-of-the-art reasoning performance with faster convergence and reduced sample needs especially on the weak models. Early-stage synergy between decomposition and reflection boosts correct sample generation, overcoming LLMs' initial cognitive limits. This work establishes a paradigm for developing LLMs that acquire advanced reasoning beyond pretraining, offering scalable solutions for complex mathematical tasks.

## REPRODUCIBILITY STATEMENT

We provide a full specification of our experimental setup in Section 5 and Appendix C, encompassing the benchmarks, training data, baseline configurations, and all hyperparameters to ensure reproducibility. The source code is publicly available at: `https://anonymous.4open.science/r/Cog-Rethinker-50C7/`.

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

## A    MATHEMATICAL DERIVATIONS

In this section, we present the proof of Theorem 1. To facilitate this, we first introduce Lemma 1.

**Lemma 1** (Stage-wise Iteration Complexity). *Under the same assumptions as Theorem 1, for any target accuracy $\epsilon > 0$ and any $\delta \in (0, 1)$, with probability at least $1 - \delta$ the iteration complexity of stage $m \in \{\text{DR}, \text{DecR}, \text{RefR}\}$ satisfies*

$$T_m(\epsilon) \leq \frac{C\sigma_m^2}{\rho_m^2 \epsilon} \log\left(\frac{1}{\delta}\right),$$

*where $C$ is a universal constant depending only on the smoothness $L$ and initial step-size $\alpha_0$. Consequently,*

$$T_{\text{RefR}}(\epsilon) < T_{\text{DecR}}(\epsilon) < T_{\text{DR}}(\epsilon).$$

*Proof.* Fix a stage $m$ and denote $J_m(\theta) = \mathbb{E}_{\tau \sim \pi_\theta}[R_m(\tau)]$. We first establish a generic bound on $\|\theta_T - \theta^*\|^2$ and then translate it into an iteration-complexity statement.

By $L$-smoothness of $J_m(\theta)$ and the update rule $\theta_{t+1} = \theta_t + \alpha_t g_m(\theta_t)$,

$$\|\theta_{t+1} - \theta^*\|^2 \leq \|\theta_t - \theta^*\|^2 + 2\alpha_t \langle g_m(\theta_t), \theta_t - \theta^* \rangle + \alpha_t^2 \|g_m(\theta_t)\|^2.$$

Taming expectation w.r.t. the randomness of $g_m$,

$$\mathbb{E}[\|\theta_{t+1} - \theta^*\|^2] \leq \mathbb{E}[\|\theta_t - \theta^*\|^2] + 2\alpha_t \langle \nabla J_m(\theta_t), \theta_t - \theta^* \rangle + \alpha_t^2 \left(\|\nabla J_m(\theta_t)\|^2 + \frac{\sigma_m^2}{N_m}\right)$$

$$\leq \mathbb{E}[\|\theta_t - \theta^*\|^2] - 2\alpha_t \mu_m \mathbb{E}[J_m^* - J_m(\theta_t)] + \alpha_t^2 \left(2L(J_m^* - J_m(\theta_t)) + \frac{\sigma_m^2}{N_m}\right),$$

where the last inequality uses the Polyak-Łojasiewicz (PL) Condition $\|\nabla J_m(\theta)\|^2 \geq 2\mu_m(J_m^* - J_m(\theta))$ and smoothness $J_m^* - J_m(\theta) \geq \frac{1}{2L}\|\nabla J_m(\theta)\|^2$.

Let $\Delta_t = \mathbb{E}[J_m^* - J_m(\theta_t)]$. Then

$$\mathbb{E}[\|\theta_{t+1} - \theta^*\|^2] \leq (1 - 2\alpha_t \mu_m + 2L\alpha_t^2)\mathbb{E}[\|\theta_t - \theta^*\|^2] + \alpha_t^2 \frac{\sigma_m^2}{N_m}.$$

Choosing $\alpha_t = \frac{1}{\mu_m(t+1)}$ yields

$$\mathbb{E}[\|\theta_T - \theta^*\|^2] \leq \frac{C\sigma_m^2}{\mu_m^2 N_m T} \leq \frac{C'\sigma_m^2}{\rho_m^4 \mu_{\min}^4 N_m T},$$

where we used $\mu_m = \rho_m^2 \mu_{\min}^2 / 2$.

To achieve $\mathbb{E}[\|\theta_T - \theta^*\|^2] \leq \epsilon$, it suffices to tame

$$T \geq \frac{C\sigma_m^2}{\rho_m^2 \epsilon} \log\left(\frac{1}{\delta}\right).$$

High-probability extension follows from Azuma-Hoeffding applied to the martingale sequence $M_t = \|\theta_t - \theta^*\|^2 - \mathbb{E}[\|\theta_t - \theta^*\|^2]$.

Inserting the empirical inequalities

$$\rho_{\text{RefR}} > \rho_{\text{DecR}} > \rho_{\text{DR}}, \qquad \sigma_{\text{RefR}}^2 < \sigma_{\text{DecR}}^2 < \sigma_{\text{DR}}^2$$

into the bound gives

$$T_{\text{RefR}}(\epsilon) < T_{\text{DecR}}(\epsilon) < T_{\text{DR}}(\epsilon),$$

which completes the proof. $\qquad\qquad\qquad\qquad\qquad\qquad\qquad\qquad\qquad\qquad\qquad\qquad\quad\square$

Then, we can conduct the proof of Theorem 1.

**Theorem 1** (Convergence Rate across Stages (Restatement)). *Let $m \in \{\text{DR}, \text{DecR}, \text{RefR}\}$ index the rollout stages of Cog-Rethinker. Assume (i) horizon $H$ is finite and rewards $r_m(\cdot, \cdot) \in [0, 1]$; (ii) for DecR, the problem is decomposed into sub-problems of horizon $H' < H$; (iii) for RefR, reflection is performed on a sub-tree of horizon $H'' \leq \gamma H'$ with $\gamma \in (0, 1)$. Then the policy-gradient estimator,*

$$g_m(\theta) = \sum_{t=0}^{H_m - 1} \nabla_\theta \log \pi_\theta(a_t | q_t) G_{t,k}, \quad G_{t,m} = \sum_{u=t}^{H_m - 1} r_m(q_u, a_u)$$

*satisfies*

$$\mathrm{Var}(g_{\mathrm{RefR}}) \;\leq\; \gamma(1-\eta)\,\mathrm{Var}(g_{\mathrm{DecR}}) \;<\; \mathrm{Var}(g_{\mathrm{DecR}}) \;<\; \mathrm{Var}(g_{\mathrm{DR}}),$$

*where* $\eta \in (0,1)$ *is the variance-reduction factor induced by importance-sampling the error sub-tree,* $\mathrm{Var}(\cdot)$ *represents the variance Consequently, for target accuracy* $\epsilon > 0$,

$$T_{\mathrm{RefR}}(\epsilon) \;<\; T_{\mathrm{DecR}}(\epsilon) \;<\; T_{\mathrm{DR}}(\epsilon).$$

*where* $T(\cdot)$ *represents the iteration complexity.*

*Proof.* We bound $\mathrm{Var}(g_m)$ for each $m$ by analysing the reward-to-go variance.

From Sutton et al. (1999),

$$\mathrm{Var}(g_m) = \mathbb{E}\left[ \sum_{t=0}^{H_m-1} \left\| \nabla_\theta \log \pi_\theta(a_t|s_t) \right\|^2 \mathrm{Var}_t(G_{t,m}) \right],$$

where $H_m = H, H', H''$ for DR, DecR, RefR respectively and

$$\mathrm{Var}_t(G_{t,m}) = \mathbb{E}\left[ \left( \sum_{u=t}^{H_m-1} r_m(s_u,a_u) - Q_m(s_t,a_t) \right)^2 \;\Big|\; s_t, a_t \right].$$

Since rewards are in $[0,1]$ and $H_m = H$,

$$\mathrm{Var}_t(G_{t,\mathrm{DR}}) \leq (H-t)^2 \leq H^2.$$

Hence,

$$\mathrm{Var}(g_{\mathrm{DR}}) \leq C\,H^3,$$

where $C = \max_t \mathbb{E}\|\nabla_\theta \log \pi_\theta(a_t|s_t)\|^2$.

Decomposition splits the original MDP into $k$ sub-MDPs each of horizon $H' = H/k$. For any sub-problem $i$,

$$\mathrm{Var}_t(G^{(i)}_{t,\mathrm{DecR}}) \leq (H')^2 = H^2/k^2.$$

Summing over $k$ sub-problems,

$$\mathrm{Var}(g_{\mathrm{DecR}}) \leq C\,k\,(H')^3 = C\,H^3/k^2 \;<\; \mathrm{Var}(g_{\mathrm{DR}}).$$

Reflection only resamples a *sub-tree* of relative size $\gamma \in (0,1)$ and uses importance weight $w \leq 1$ on the erroneous part. The law of total variance gives

$$\mathrm{Var}_t(G_{t,\mathrm{RefR}}) = \mathbb{E}[\mathrm{Var}_t(G_{t,\mathrm{RefR}} \mid \text{sub-tree})] + \mathrm{Var}_t(\mathbb{E}[G_{t,\mathrm{RefR}} \mid \text{sub-tree})].$$

The first term is bounded by $\gamma(H')^2$; the second term is reduced by the *negative-curriculum* effect (mnowing the wrong path) and satisfies

$$\mathrm{Var}_t(\mathbb{E}[G_{t,\mathrm{RefR}} \mid \text{sub-tree}]) \leq (1-\eta)\mathrm{Var}_t(G_{t,\mathrm{DecR}})$$

with $\eta \in (0,1)$ depending on the overlap between wrong and corrected trajectories. Thus

$$\mathrm{Var}(g_{\mathrm{RefR}}) \leq C\,\gamma(1-\eta)\,H^3/k^2 \;<\; \mathrm{Var}(g_{\mathrm{DecR}}).$$

From smoothness and PL Condition (as in Lemma 1),

$$\mathbb{E}[\|\theta_T - \theta^*\|^2] \leq \frac{C'\mathrm{Var}(g_m)}{T}.$$

Therefore the iteration complexity

$$T_m(\epsilon) \leq \frac{C'\mathrm{Var}(g_m)}{\epsilon}$$

satisfies the ordering

$$T_{\mathrm{RefR}}(\epsilon) \;<\; T_{\mathrm{DecR}}(\epsilon) \;<\; T_{\mathrm{DR}}(\epsilon). \qquad \square$$

## B  TRAINING DETAILS

In this section, we present the training details of our Cog-Rethinker, including the training algorithm, prompt templates and implementation details.

### B.1  TRAINING ALGORITHM

To better understand the training procedure of our Cog-Rethinker, we present the pseudo code in Algorithm 1.

---

**Algorithm 1** The training pipeline of our Cog-Rethinker.

---

**Require:** Initial policy $\pi_\theta$; reward model $R$; problems data $\mathcal{D}$; hyperparameters $\varepsilon_{\text{low}}$, $\varepsilon_{\text{high}}$, $\lambda$
1: **for** step $= 1, \ldots, M$ **do**
2:     Sample a batch $\mathcal{D}_b$ from $\mathcal{D}$
3:     Update the old policy model $\pi_{\theta_{old}} \leftarrow \pi_\theta$
4:     Sample $G$ outputs $\{o_i\}_{i=1}^G \sim \pi_{\theta_{\text{old}}}(\cdot \mid q)$ for each problem $q \in \mathcal{D}_b$ using normal prompt template
5:     Compute rewards $\{r_i\}_{i=1}^G$ for each $o_i$ by running $R$
6:     Calculate accuracy rate $\gamma$ for each problem $q$
7:     Filter out $o_i$ and add remaining to dynamic sampling buffer
8:     **if** buffer size $n_b < N$ **then**
9:         **for** each $q$ with $\gamma = 0$ **do**
10:             Sample $G$ outputs $\{o_i\}_{i=1}^G \sim \pi_{\theta_{\text{old}}}(\cdot \mid q)$ using decomposition prompt template
11:             Repeat lines 5-7
12:         **end for**
13:     **end if**
14:     **if** buffer size $n_b < N$ **then**
15:         **for** each $q$ with $\gamma = 0$ **do**
16:             Sample $G$ outputs $\{o_i\}_{i=1}^G \sim \pi_{\theta_{\text{old}}}(\cdot \mid q)$ using reflection prompt template
17:             Repeat lines 5-7
18:         **end for**
19:     **end if**
20:     **if** buffer size $n_b < N$ **then**
21:         **continue**
22:     **end if**
23:     For each $o_i$ in buffer, compute $\hat{A}_{i,t}$ for $t$-th token of $o_i$
24:     **for** iteration $= 1, \ldots, \mu$ **do**
25:         Update $\pi_\theta$ by minimizing objective in Eq. (9)
26:     **end for**
27: **end for**
**Ensure:** Optimized policy $\pi_\theta$

---

### B.2  PROMPT TEMPLATES

As shown in Figure 2 of our Cog-Rethinker, there are two new rollout procedures in the rollout stage of RL training. For the decomposition rollout procedure, we present the details of full prompt template in the following.

### B.3  IMPLEMENTATION DETAILS

Our Cog-Rethinker is easy to implement. We present the training algorithm in Algorithm 1, with all training procedures based on VeRL (Sheng et al., 2024). In the practical training procedure, we introduce a hyperparameter $\lambda$ to control the Gaussian regularization strength. Our experiments are conducted on $8 \times$ NVIDIA A800 GPUs, with $\lambda$ set to 0.04. Additional results on hyperparameter analysis are presented in Section C.3.

---

**Prompt Template for Decomposition Rollout**

Solve the following math problem step by step. The last line of your response should be of the form Answer: $Answer (without quotes) where $Answer is the answer to the problem. Let's attempt a subproblem decomposition approach:

1. Split the original problem into smaller, logically related subproblems that will assist you in solving the original problem-quantity depends on the problem's logic and your expertise.

2. Address each subproblem individually, analyzing the reasoning behind your solutions.

3. Combine the subproblem solutions to tackle the original, more complex problem.

**Example problem:** Solve the equation $\frac{3(x-2)}{4} - \frac{2x+5}{3} = \frac{1}{6}$.

**Solution:**

**Subproblem 1:** Eliminate denominators by multiplying all terms by the least common multiple (LCM) of 4, 3, and 6, which is 12: $12 \cdot \frac{3(x-2)}{4} - 12 \cdot \frac{2x+5}{3} = 12 \cdot \frac{1}{6}$. Simplifies to: $9(x-2) - 4(2x+5) = 2$.

**Subproblem 2:** Expand and simplify: $9x - 18 - 8x - 20 = 2$. Combine like terms: $x - 38 = 2$

**Subproblem 3:** Isolate the variable: $x = 2 + 38, x = 40$.

**Final Solution:**

Substituting $x = 40$ back into the original equation confirms both sides equal $\frac{1}{6}$.

Answer: 40

Remember to put your answer on its own line after "Answer:"

---

**Prompt Template for Reflection Rollout**

Solve the following math problem step by step. The last line of your response should be of the form Answer: $Answer (without quotes) where $Answer is the answer to the problem. Let's attempt a subproblem decomposition approach:

1. Analyze the problem. Read the problem carefully and clarify the known conditions and final requirements.

2. Identify the error. Locate the error type in the existing solution (concept error, calculation error, logical loophole).

3. Correct step by step. Correct the error step by step, retain the reasonable part of the original solution and correct the error point one by one.

4. Verify the answer. Use multiple methods to verify the correctness of the final answer.

**Problem:** Solve the system of equations:

1) $2x + 3y = 7$

2) $4x - y = 3$

**The existing wrong solution:**

**Subproblem 1:** Solve equation 2 for y. Starting equation: $4x - y = 3$.

**Subproblem 2:** Substitute into equation 1. Correct substitution should be: $2x + 3(4x - 3) = 7$.

**Subproblem 3:** Solve the simplified equation. Equation being solved: $2x + 12x = 7$.

**Subproblem 4:** Find corresponding y value. Using partial solution: $y = 4(0.5) = 2$.

Answer: $(0.5, 2)$

Remember to put your answer on its own line after "Answer:"

---

## C  ADDITIONAL EXPERIMENTS

### C.1  TRAINING VISUALIZATION

In addition to the results in Figure 4, we conduct further experiments comparing our Cog-Rethinker with DAPO, with the results presented in Figure 6. Specifically, we use maj@16 as the comparison metric. Figure 6 shows our Cog-Rethinker consistently outperforms DAPO in all tests. On the MATH500 benchmark in Figure 6(a), our method outperforms 2% than DAPO and converges faster, becoming stable after 150 training steps. For GPQA-Diamond in Figure 6(b), our method maintains

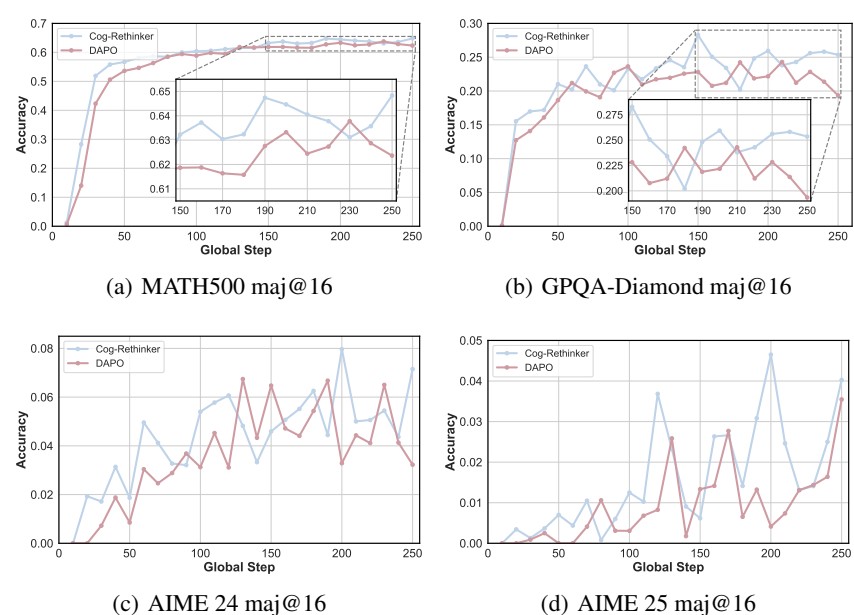

(a) MATH500 maj@16

(b) GPQA-Diamond maj@16

(c) AIME 24 maj@16

(d) AIME 25 maj@16

Figure 6: Additional training visualization between our Cog-Rethinker with DAPO.

accuracy between 22.5% and 25% after step 200, and shows more stable performance between 210-250 steps than DAPO. The smaller graphs show our method stays stable during important training periods (steps 150-250), while DAPO stops improving or gets worse.

The AIME results in Figure 6(c)-(d) show both methods have unstable results due to difficult problems, but our approach works more reliably. Specially, for AIME 25, our method keeps a 2% to 4% lead even when results vary more.

## C.2 HYPERPARAMETER ANALYSIS

In our experiments, the main hyperparameters are $\varepsilon_{\text{low}}$ and $\varepsilon_{\text{high}}$, which are used in DAPO for clip-higher. We adopt the default DAPO settings of $\varepsilon_{\text{low}} = 0.20$ and $\varepsilon_{\text{high}} = 0.28$.

For our newly introduced hyperparameter $\lambda$, we analyze its influence by testing values in $\{0.04, 0.02, 0.01, 0.005\}$, with results shown in Figure 7. The results demonstrate dataset-dependent responses to SFT coefficient $\lambda$ variations. GSM8K and MATH-500 show $\lambda$-sensitivity. AMC 2023, Gaokao 2023en exhibit limited accuracy variation (42.5%-47.5%) across $\lambda$ values. Challenging benchmarks, such as AIME 2024/2025, OlympiadBench, maintain consistently low performance (3.3%-6.7%), showing minimal $\lambda$-sensitivity. Minerva and GQA-Diamond display moderate accuracy (15.1%-24.4%). These patterns indicate that $\lambda$ tuning primarily benefits already well-performing datasets, while complex reasoning tasks require fundamental model improvements beyond hyperparameter optimization. The findings highlight the critical interplay between dataset characteristics and regularization effectiveness in mathematical reasoning tasks.

Regarding optimization steps per global step, the main experiments use a batch size of 32 with four optimization steps. We also conduct experiments with mini batch sizes 64 and 128, corresponding to two and one optimization steps respectively. The results are shown in Figure 8.

Figure 8 evaluates three mini-batch configurations across multiple benchmarks, demonstrating consistent performance advantages for smaller mini batch sizes. The results show a clear hierarchy where batch_size=32 outperforms larger batches in most tasks, achieving 77.5% on GSM8K (vs. 74.8% for 128) and showing the most substantial 5.6% gain on MATH-500. While this trend holds universally, the magnitude varies by domain - mathematical problems like MATH-500 benefit most, while complex tasks like AIME 25 show greater variance (6.7% for 32 vs. 0% for 64). The find-

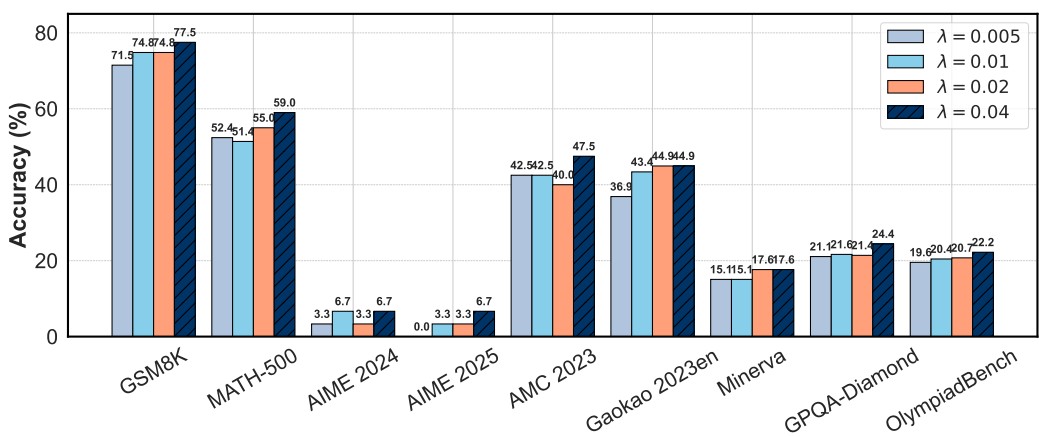

Figure 7: Performance comparison of different SFT coefficient $\lambda$ for policy $\pi_\theta$ update.

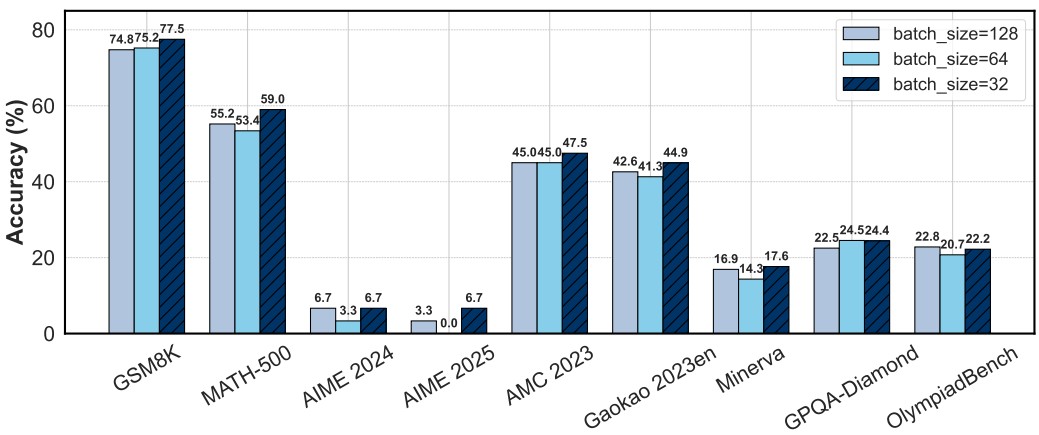

Figure 8: Performance comparison of different training mini batch size for policy $\pi_\theta$ update, where batch_size=128, batch_size=64 and batch_size=32 represent the optimization steps as 1, 2 and 4 in each global step, respectively.

ings confirm batch size selection should balance computational efficiency with these task-specific patterns, particularly for mathematical problems where smaller batches show clearest advantages.

## C.3 PERFORMANCE ANALYSIS

We conduct the experiments on Llama3.2 (Meta AI Team, 2024) in Table 3. The results also demonstrate the superiority of our proposed Cog-Rethinker, particularly in scenarios where the baseline model Llama exhibits limited reasoning capability. Both BODF and DAPO failed to complete the training process due to challenges associated with online filter design. Specifically, Llama's weak mathematical reasoning ability led to a substantial number of samples with zero accuracy, resulting in insufficient valid training batches. In contrast, our Cog-Rethinker successfully completed training and demonstrated superior performance compared to baseline methods, which can be attributed to its effective decomposition and reflection rollout mechanism.

We also conducted comparison of our Cog-Rethinker and DAPO in handling negative samples under the same rollout time in Table 4. In extreme cases, our Cog-Rethinker requires up to three times the number of rollouts when performing both decomposition and reflection operations. To ensure a fair comparison, we allocated the maximum potential number of rollouts on all samples to DAPO (DAPO_rollout). While increasing the number of rollouts led to a minor performance improvement

Table 3: Overall accuracy performance of Llama3.2 models on various reasoning benchmarks. The best and second best results are in **bold** and underlined.

| Method | GSM8K | MATH-500 | AIME 24 | AIME 25 | AMC 2023 | Gaokao 2023en | Minerva | Olympiad |
|---|---|---|---|---|---|---|---|---|
| Llama3.2-1B-Base | 1.74 | 3.80 | 0.00 | 0.00 | 0.00 | 0.00 | 2.57 | 1.04 |
| PPO | 29.09 | 15.00 | 0.00 | 0.00 | 2.50 | 2.60 | 2.21 | 3.41 |
| GRPO | 28.60 | 13.41 | 0.00 | 0.00 | 10.00 | 5.19 | 1.47 | 3.30 |
| Reinforce++ | **34.93** | 14.60 | 0.00 | 0.00 | 12.50 | 5.19 | 1.47 | 3.62 |
| BODF | – | – | – | – | – | – | – | – |
| DAPO | – | – | – | – | – | – | – | – |
| **Cog-rethinker** | 34.70 | **17.80** | 0.00 | 0.00 | **18.50** | **8.68** | **3.49** | **4.38** |
| Llama3.2-3B-Base | 6.97 | 6.40 | 0.00 | 0.00 | 0.00 | 0.00 | 5.51 | 1.48 |
| PPO | 20.43 | 17.80 | 0.00 | 0.00 | 10.00 | **17.53** | 7.78 | 5.63 |
| GRPO | 24.30 | 17.40 | 0.00 | 0.00 | **17.50** | 8.57 | 8.04 | 5.19 |
| Reinforce++ | 27.81 | 22.20 | 0.00 | 0.00 | 12.50 | **17.53** | 8.15 | 5.78 |
| BODF | – | – | – | – | – | – | – | – |
| DAPO | – | – | – | – | – | – | – | – |
| **Cog-rethinker** | **32.73** | **26.60** | 0.00 | 0.00 | **17.50** | 17.27 | **9.23** | **8.48** |

Table 4: Comparison of our Cog-Rethinker and DAPO on Qwen2.5 models in handling negative samples under the same rollout number.

| Method | GSM8K | MATH-500 | AIME 24 | AIME 25 | AMC 2023 | Gaokao 2023en | Minerva | Olympiad |
|---|---|---|---|---|---|---|---|---|
| Qwen2.5-1.5B-Base | | | | | | | | |
| DAPO | 77.56 | 56.00 | 6.67 | 0.00 | **47.50** | 42.34 | 16.54 | 22.22 |
| DAPO$_{rollout}$ | **77.78** | 55.56 | 6.67 | 0.00 | 47.00 | 43.48 | 16.77 | 22.22 |
| **Cog-Rethinker** | 77.51 | **59.00** | 6.67 | **6.67** | 47.50 | **44.94** | **17.65** | 22.22 |
| Qwen2.5-7B-Base | | | | | | | | |
| DAPO | 92.21 | 79.40 | **26.67** | 26.67 | 69.00 | 63.22 | 31.88 | 42.44 |
| DAPO$_{rollout}$ | 93.02 | 78.90 | 23.33 | 26.67 | 66.88 | 64.41 | 31.27 | 43.78 |
| **Cog-Rethinker** | **93.32** | **80.60** | **26.67** | 26.67 | **73.50** | **65.52** | **32.98** | **44.22** |

in DAPO, our Cog-Rethinker achieves significantly greater gains. This discrepancy arises because DAPO$_{rollout}$ encounters limitations inherent to the base model's capacity, and merely increasing rollout number of negative samples does not enable the model to surpass these inherent constraints. In contrast, our Cog-Rethinker enables the model to break down complex questions into simpler subproblems through its rollout mechanism, thereby transcending the base model's limitations. This observed phenomenon substantiates the effectiveness of our Cog-Rethinker.

## D    LIMITATIONS

Our Cog-Rethinker's performance is contingent on the quality of the base model and the predefined demonstrations in its metacognitive buffer, which may limit its adaptability to unseen or highly novel problems. The framework assumes access to accurate subproblem decompositions and reflections, which may not always be feasible in practice. Additionally, the binary reward system lacks granularity to reward intermediate reasoning steps, potentially hindering nuanced learning. The experiments focus on mathematical reasoning, and generalization to other domains remains untested.

## E    USE OF LLMs

We use LLMs only to refine the language and grammar in our paper. We do not use them for generating research ideas or for finding related work. We provide our complete original text to OpenAI's GPT-4o with instructions to make it more professional, coherent, and native-sounding for a research paper. We then carefully review all suggestions to guarantee that no factual content is altered and that all changes remain true to our original writing.

