# OpenReview forum: "Cog-Rethinker: Hierarchical Metacognitive Reinforcement Learning for LLM Reasoning"
_ICLR.cc/2026/Conference — ICLR 2026 Conference Withdrawn Submission_

### Official Review · Reviewer_wJBA · 2025-10-26

**Soundness:** 2
**Presentation:** 2
**Contribution:** 2
**Rating:** 2
**Confidence:** 3

**Summary:**

This paper introduces Cog-Rethinker, a framework designed to improve the sample efficiency of post-training in large language models (LLMs). For complex reasoning tasks, direct rollouts from LLMs often yield incorrect outputs, making training on these samples inefficient. To address this issue, the authors propose two additional stages: decomposition and reflection. In the decomposition rollout, a complex problem is broken down into simpler subproblems, guided by retrieved demonstrations using the BM25 retrieval algorithm. In the reflection rollout, incorrect responses are revised by leveraging prior successful rewriting experiences. Experimental results show that the proposed method significantly enhances training efficiency for Qwen-2.5 models compared to five baseline approaches.

**Strengths:**

* The experiment compares with 5 different baselines over 9 mathematical reasoning benchmarks.
* A proof is provided, showing that the convergence rate of the proposed method is better than the direct roll-out.

**Weaknesses:**

* Statistical significance tests are missing in Table 1 and Table 2. Without these tests, it may be inappropriate to highlight the best and second-best results, making it difficult to assess whether the proposed method truly outperforms the baselines.

* Model selection lacks sufficient justification. The experiments are limited to Qwen2.5-5B and Qwen2.5-7B. Although results for Llama3.2-1B and Llama3.2-3B are briefly presented in Appendix C.3, the table appears incomplete. It remains unclear whether the weak performance stems from the limited mathematical reasoning capability of the Llama models or from potential implementation issues. Consequently, it is difficult to evaluate the generalizability of the proposed approach. Additional experiments on alternative models, such as Phi-2 or Gemma, would strengthen this work.

* The connection between cognitive engineering and the proposed method requires further clarification. In its current form, the paper primarily appears to leverage chain-of-thought reasoning and self-reflection mechanisms to improve sample efficiency, rather than demonstrating a clear cognitive engineering perspective.

**Questions:**

* Unclear notation. What does $A$ stand for? In Equation 1, it is the advantage function. In L215, it is the final answer. Are they the same thing?
* IN L240, the authors mentioned they "guide" the policy to solve the subproblems. How exactly do they implement it?
* Figure 1 is unclear to me. How to interpret that the proposed method can generate more correct samples?

---

### Official Review · Reviewer_hoZy · 2025-10-30

**Soundness:** 3
**Presentation:** 3
**Contribution:** 2
**Rating:** 6
**Confidence:** 2

**Summary:**

The paper introduces Cog-Rethinker, a novel framework for improving LLM reasoning through Hierarchical Metacognitive Reinforcement Learning. Standard RL approaches often struggle with weak LLMs because complex reasoning tasks require numerous samples to find a valid solution.
Cog-Rethinker addresses this by adopting a metacognitive rollout mechanism that breaks down a complex problem into a sequence of simpler, more manageable sub-problems. By solving these sub-problems hierarchically, the model effectively bypasses the inherent capacity constraints of the base LLM, leading to better inferential performance. The experiments focus on improving mathematical reasoning capabilities.

**Strengths:**

* This paper introduces a novel hierarchical metacognitive design that mimics human problem-solving strategies.


* The paper reports result across several benchmarks and performs ablation studies showing the contribution of each module.

**Weaknesses:**

* Experiments focus almost exclusively on mathematical reasoning. Generalization to other reasoning-heavy domains (e.g.,  commonsense reasoning) remains unproven.


* The framework’s weakness lies in its dependence on prior knowledge and input quality, making it less adaptable to novel problems and sensitive to initial setup.

* The approach requires multiple rollouts (direct, decomposition, reflection) for a single problem, which increases computational cost per training step. The paper argues for superior sample efficiency, but a direct discussion of the total computational cost compared to baselines is missing.

* The use of a simple binary (correct/incorrect) reward function, while praised for stability, lacks granularity. It cannot reward partially correct reasoning or high-quality intermediate steps, which may limit the potential for learning more nuanced reasoning strategies.

**Questions:**

Please refer to "Weaknesses".

---

### Official Review · Reviewer_v1rL · 2025-10-31

**Soundness:** 2
**Presentation:** 2
**Contribution:** 2
**Rating:** 2
**Confidence:** 4

**Summary:**

Cog-Rethinker proposes a hierarchical metacognitive reinforcement learning framework that enhances sample efficiency and reasoning capability through a three-stage rollout process (direct, decomposition, reflection), combined with supervised fine-tuning (SFT) to ensure train-test consistency and enable cold-start learning.

**Strengths:**

1. Instead of discarding problems that receive zero accuracy in the rollout such as DAPO, the paper improves utilization of negative samples.

2. Cog-Rethinker is especially effective for base models with limited reasoning capacity.

**Weaknesses:**

1. Cog-Rethinker is presented as a plug-and-play framework that could be combined with various RL algorithms (e.g., PPO, GRPO, Reinforce++). However, the paper only integrates it with DAPO.

2. The decomposition strategy resembles in-context learning, where the model may merely mimic surface-level patterns from provided templates rather than genuinely mastering systematic problem decomposition. Although the authors claim to enhance “diversity” through dynamic buffer updates, it remains questionable whether similarity-based retrieval (e.g., BM25) combined with fixed decomposition prompts can truly promote diversity in reasoning steps.

3. The paper aligns outputs from three distinct rollout prompt templates by converting them into the direct-prompt format and applying SFT. While this ensures train-test consistency, it may discard valuable metacognitive signals. Are there more principled alternatives that could better preserve the benefits of hierarchical reasoning and RL?

4. The paper frequently references “metacognition” and “cognitive engineering”, but the proposed method does not explicitly model or validate any cognitive mechanisms.

5. The experiments focus solely on mathematical reasoning benchmarks. To better demonstrate the generality of Cog-Rethinker, evaluations on broader QA or open-ended reasoning tasks  would be more convincing—especially since the base models used (Qwen base) are general-purpose, not math-specialized.

6. The rule-based reward function in the paper only checks the correctness of the final answer but does not consider factors such as format and logical consistency. The simple reward signal leads to a more serious problem of model reward hacking.

**Questions:**

Please see weaknesses.

---

### Official Review · Reviewer_HtjF · 2025-11-01

**Soundness:** 2
**Presentation:** 3
**Contribution:** 2
**Rating:** 4
**Confidence:** 3

**Summary:**

Cog-Rethinker is a hierarchical metacognitive reinforcement learning framework for LLM reasoning. It leverages decomposition-based demonstrations and corresponding chain-of-thought strategies to tackle complex problems. To address the inconsistency issues introduced by different prompt templates, Cog-Rethinker integrates SFT loss with DAPO loss. Experiments on various mathematical reasoning benchmarks demonstrate the effectiveness of the the method.

**Strengths:**

- The motivation makes sense.

- The paper is well-written.

- The idea of Cog-Rethinker is interesting, and the paper conducts extensive experiments to verify its effectiveness.

**Weaknesses:**

1. Baseline: An important baseline is missing — one that uses SFT as a cold start. Before applying RL, you could perform SFT on the decomposition and reflection examples to inject the CoT strategies (decomposition and reflection) described in your paper.

2. Base Model: The paper conducts extensive experiments on Qwen2.5-1.5B-Base, Qwen2.5-7B-Base, and Llama3.2-3B-Base. However, these are not the latest models. I would like to see how Cog-Rethinker performs on newer or larger models, such as Qwen3.

3. Limited evaluation domains. All experiments are conducted on mathematical reasoning benchmarks. It would be better to confirm the effectiveness in other domains, such as coding or commonsense reasoning.

**Questions:**

1. Can the DAPO baseline prompt include decomposition and reflection hints or demonstrations? If so, DAPO might be able to learn these chain-of-thought strategies through pure reinforcement learning directly.

2. Why don’t you use SFT as a cold start to inject these chain-of-thought strategies (such as decomposition and reflection)?

3. Can you verify your conclusions on larger Qwen2.5 models and more recent versions, such as Qwen3?

4. Could you conduct additional experiments to perform sensitivity analysis on the hyperparameter lambda?

---

### Note · Authors · 2025-12-26

I have read and agree with the venue's withdrawal policy on behalf of myself and my co-authors.